# Experimental Analysis of Passive Strategies in Houses with Glass Façades for the Use of Natural Light

Patricia Aguilera-Benito [1],* , Carolina Piña-Ramírez [2] and Sheila Varela-Lujan [3]

1   Departamento de Tecnología de la Edificación, Escuela Técnica Superior de Edificación, Universidad Politécnica de Madrid, 28040 Madrid, Spain
2   Departamento de Construcciones Arquitectónicas y su Control, Escuela Técnica Superior de Edificación, Universidad Politécnica de Madrid, 28040 Madrid, Spain; carolina.pina@upm.es
3   Escuela Técnica Superior de Edificación, Universidad Politécnica de Madrid, 28040 Madrid, Spain; sheila.varela.lujan@alumnos.upm.es
*   Correspondence: patricia.aguilera@upm.es

**Abstract:** The main objective of this research was to analyze the passive solutions that help to reduce energy consumption through the use of natural light in buildings. In this case, the analysis focused on the use of natural light in buildings with all façades made of glass. Buildings designed with the criterion of regenerative sustainability, such as glass buildings, consider not only energy efficiency and the use of renewable and/or alternative energies, but also the improvement of the health and well-being of users—very important factors during an era in which the time spent at home has been increased due to the COVID-19 pandemic (disease caused by the SARS-CoV-2 virus). This experimental research was based on the analysis of data collected during a full year of monitoring of two scale models of Farnsworth House, a house with glass façades in all of its orientations. The results allowed us to conclude that these passive elements must be taken into account in the initial design of a building; a solar control sheet is also advisable in this type of glass building because it greatly reduces the illuminance inside the spaces, and the use of overhangs distributes the illuminance inside the rooms more evenly. In addition, the use of scale models provides a more accurate view of the parameters related to light energy.

**Keywords:** light energy; passive strategies; use of natural light; solar control sheet; glass façades



## 1. Introduction

Currently, the construction sector generates a significant environmental impact due to its energy consumption and the large amount of pollutants emitted into the atmosphere throughout the entire life cycle of buildings. Therefore, the European Union has set guidelines aimed at improving their energy efficiency, adding the obligation to build buildings with "almost zero energy" [1]. One of the main objectives established is to increase the use of alternative energy, such as solar energy, a free and inexhaustible source of energy on Earth. Solar radiation is a source of energy savings, both by reducing electricity consumption by artificial lighting and by saving money on heat generation facilities, as well as improving human well-being [2].

In Spain, buildings are responsible for more than 30% of primary energy consumption and 28% of $CO_2$ emissions into the atmosphere. Furthermore, lighting installations in Spain account for 20% of the total energy consumption; this percentage is considered high due to the fact that many traditional light bulbs, fluorescent lamps, and mercury vapor lamps are still installed [3–5]. For all of these reasons, architecture has evolved to reduce this impact by incorporating previous design aspects that consider natural lighting as a criterion to achieve a reduction of up to 50% in the total energy consumption, thus bringing us closer to buildings of almost zero consumption and a sustainable architecture—that is to say, buildings that generate zero impact on the environment. However, it has been

confirmed that these design criteria are not sufficient if we want to stop climate change and improve people's quality of life. The design of buildings must move toward the construction of houses that generate positive impacts, at both environmental and societal levels. Buildings designed with the criterion of regenerative sustainability consider, in addition to energy efficiency and the use of renewable and/or alternative energies, the improvement in the health and well-being of users; among these buildings are houses with large glazed surfaces that allow integration with the environment and an improvement in health due to the provision of natural light [6–8]. In addition, in this sense, bioclimatic architecture is a benchmark on which to consolidate this new concept, since it considers the use of not only the sun as an alternative energy source, but also sunlight as a source of health-related and other benefits for human beings [9–11]. Glass façades benefit from both heat and light energy, provided by the sun, as a renewable energy source that does not need any equipment or system to be transformed from primary to final energy.

In recent years, glass façades have become the standard for large buildings for non-residential use, and this type of contemporary architecture has become one of the hallmarks of cities [12]. The glass enclosure has its origins as a solution to the specific needs present in unique buildings; these needs are associated with the more technological aspects of luminosity and transparency [13]. Different companies or brands seek exclusivity in the design of their corporate buildings (flagship), and for this reason, they seek buildings with unique architecture to maintain the image of the firm. All of these conditions are difficult to see transferred to residential buildings because glass in homes is primarily only used for stairwells or terrace enclosures [14], but glass buildings benefit from sunlight, both as an alternative energy source and as a source of health-related benefits [15,16].

With regard to the studies found on the light behavior in glass buildings, it should be noted that there are numerous investigations that show the benefit of natural light for the well-being and comfort of occupants. Being able to work in natural light improves people's performance, as well as mood, and contributes to a better quality of life [17–19]. Many studies have focused on office buildings, but few have shown results for residential buildings, where—in the context of the pandemic—the number of hours spent in the home has increased due to teleworking and online classes [20–26].

Additionally, taking into account the design of the building, latitude and climate are two important and highly influential factors for natural lighting systems and for the installation of protection. Lower latitudes (around 25° to 40° north and south) have the highest sunshine duration values, contrary to the highest latitudes (above 50° north and south). In addition to having low values of annual sunlight duration, they are affected by high winds and have a much more cloudy, rainy, and unstable climate. Another important point is the passive strategies that are designed to optimize lighting levels to the maximum and to mitigate high lighting levels. Some of these strategies are slat systems that can control the incidence of solar rays inside spaces or the use of curtains of different fabrics or venetian types [27–32]. An important point in buildings with large glass panes is the possibility of capturing all of the necessary natural light in order to minimize the costs of artificial lighting [33–36].

The evolution of glass and passive solutions can make the construction of this type of building viable in Spain, which experiences long periods of sunshine. For this reason, it is necessary to investigate passive strategies that minimize this negative impact (higher luminous energy consumption) on the increase in the energy demand of the building, avoiding any detrimental effect on sustainability, the health of the occupants, and the environment. This research is based on a partial approach to energy savings in a fully glazed building. It focuses specifically on the analysis of daylighting in this type of unique building. This study is complemented by the research "Thermal Behavior in Glass Houses through the Analysis of Scale Models" [37], which analyzed the thermal parameters.

The following section will describe, on the one hand, the house selected as a reference for the execution of the scale models. A comprehensive description of these scale models and of the passive strategies to be analyzed is provided. Next, the monitoring carried out

and the sensors used are explained. Finally, the parameters to be analyzed in order to compare the different passive strategies in this type of housing are indicated.

## 2. Methods

In this section, the monitoring carried out on the scale models for which experimental analyses were carried out is described, as well as the technical specifications of the glass of the different façades and of the overhangs and the measured parameters.

### 2.1. Selected Home and Execution of Scale Models

The home selected for the study was Farnsworth House, which is located in the city of Plano, in the State of Illinois (United States). The house is rectangular in shape, rises 1.6 m from the ground, and has a surface area of 198.2 m$^2$, of which 59 m$^2$ correspond to the west-facing access porch. Its interior is open-plan; it only has a central volume for toilets and a facility room. The interior height is 2.9 m [38–41].

The latitude of Farnsworth House is 41° north, and the scale models were located inside the complex of the Higher Technical Building School of the Polytechnic University of Madrid (Spain), which has a latitude of 40° north. Therefore, comparing the geographical coordinates, the original dwelling located in Illinois differs in latitude by only 1° north with respect to the positioning of the scale models in Madrid. In this way, the glass building was analyzed in Madrid within the Mediterranean—continental climatic domain provided by its location in the interior of the Iberian Peninsula (Figure 1).

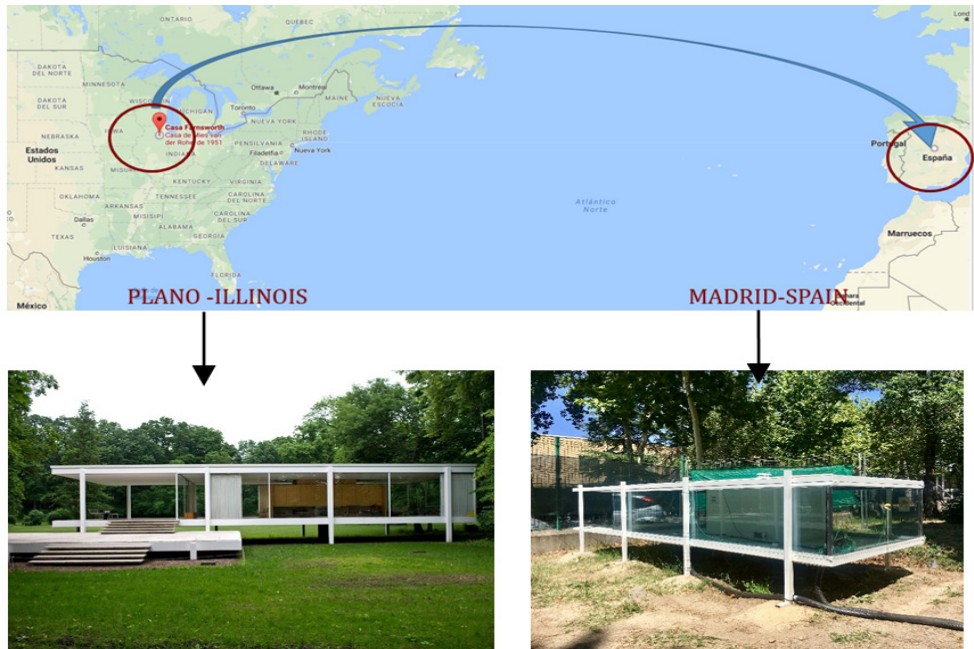

**Figure 1.** Location of the original building (Illinois) and situation of the scale models (Spain).

The scale models were separated by 2.00 m in order to avoid interference from their own shadows. In addition, the shade provided by nearby deciduous trees was studied. The following results were obtained from the analyses, where the scale models were shaded from dawn until a specified time (Table 1).

**Table 1.** Hours in shadow for scale models.

| Month | Reference Model [*] (From Dawn to:) | Improved Model [*] (From Dawn to:) |
|---|---|---|
| March and September | 6:30 a.m. | 9:00 a.m. |
| April and August | 7:30 a.m. | 9:45 a.m. |
| May and July | 8:30 a.m. | 10:15 a.m. |
| June | 9:30 a.m. | 10:30 a.m. |

(*) The construction characteristics of each scale model are explained in the following paragraphs.

As can be seen in the table above, the time difference between the beginning of the shaded period and the end, between one model and another, was around two hours. Furthermore, from 10:30 a.m. onwards, no model received shade, so the midday time slot was not affected, which was important for this analysis. For this reason, these data were taken as a reference to make the appropriate comparisons in the analyses and, thus allowing for accurate analysis of the results obtained.

After an analysis of the consulted bibliography, it was decided to run two 1/6 scale models—one of them was called the reference model and the other, the improved model, thus making it possible to compare results, knowing that they would have the same weather conditions [42–45]. Both scale models were characterized as having the same geometry and the same construction properties. The structure of the scale models was made from metal frames and pillars, the roof and floor slabs were solved with a sandwich panel with the same thermal transmittance as the original house, and the glass used for the housing in both models was placed on metal profiles, without having direct contact with them, because they were placed on neoprene bands that isolated the thermal bridge between the glass and the metal. Both models differed in the passive strategy implemented (or not) in their different orientations in order to, in this way, be able to compare the results with equivalent climatic conditions.

The passive strategies to be analyzed in the scale models, on the one hand, were the characteristics of the glazing—the reference model had a 6 mm monolithic glazing (clear float glass) in all its orientations, while the improved model had two types of glass, namely 6 + 16 + 6 mm double glazing (conventional DA) installed only on the north façade and 6 + 16 + 6 mm double glazing with LCS on face 2 of the glazing (SunGuard Solar Neutral 67), located in the rest of the orientations (south, east, and west) (Table 2).

**Table 2.** Technical characteristics of the installed glass. UNE-EN 410 (Spanish Association for Standardisation). Glass for buildings.

| Properties/Features | 6 mm Single Glass (Clear Float Class) Reference Model | Double Glass (Conventional DA) Improved Model | Double Glass + SCL (SunGuard Solar Neutral 67) Improved Model |
|---|---|---|---|
| Total solar transmission SHGC "g" (%) | 86.0 | 77.1 | 58.8 |
| Direct solar transmission (%) | 78.0 | 72.2 | 53.1 |
| Light transmission (%) | 89.0 | 81.9 | 61.0 |
| U-value (ISO 10292) (W/m$^2$ K) | 5.7 | 2.7 | 2.6 |

The second passive strategy installed for analysis during the summer months was the installation in the south and east orientations of a 30 cm cantilever at a 1/6 scale. This cantilever was the result of the calculation for 21 April at 2:00 p.m. (official time) for the south façades, corresponding to an angle of incidence of solar radiation with a horizontal of 61° (Figure 2).

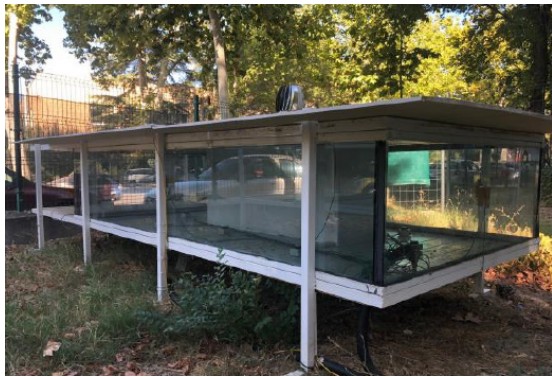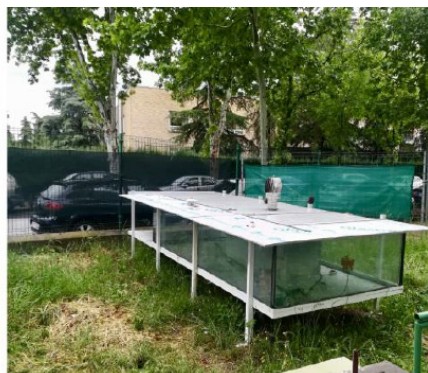

**Figure 2.** Scale models with cantilevers facing south and east.

After a previous phase of computational simulation with the Design Builder program of numerous scenarios with different configurations of strategies, both passive strategies were chosen because they offered the most representative results.

*2.2. Test Conditions*

The scale models were adapted for the monitoring to be carried out with the chosen measurement equipment, bearing in mind the objectives to be achieved, all in accordance with the applicable regulations, the guidelines of the manufacturers, and research published to date. The evaluation of the scale models was performed under a natural sky vault, and both models were measured simultaneously, so comparisons were made under the same sky conditions.

Prior to the monitoring phase of the scale models, an analysis of the behavior of the instruments to be used was carried out, i.e., eight DeltaOhm HD2021T lux meters (made by Delta Ohm Srl, Caselle di Selvazzano Padova, Italy) in each of the scale models. The lux meters were located inside the dwelling on the lower floor. This floor slab was at a height of 30 cm from the outside floor.

In addition, two Kypp & Zonen CMP3 pyranometers (made by OTT HydroMet B.V., Delft, The Netherlands) were available, which were positioned on the deck of each of the scale models. The pyranometers were located on the upper slab 90 cm above the outside floor (Figure 3).

The established nomenclature is that the first letter refers to the model in which the sensor is located (reference model—R—or modified model—M). The next three letters refer to the parameter that the sensor measures (lux or rad) and, finally, the number indicates the positioning of the sensor within each of the scale models.

Regarding the monitoring procedure, in order to validate the research and to obtain data in all seasons of the year, it was decided to monitor the scale models for one full year, which ran from August 2017 to July 2018. In the summer of 2017, the models did not have overhangs as an additional passive strategy, while during the summer months of 2018, they had overhangs installed in the south and east orientations.

The monitoring was carried out in all seasons of the year, under all weather conditions that occurred. The readings were collected in the data-logger every 2 min and were subsequently analyzed every 10 min to obtain an average value.

In addition to the sensors that were installed inside the scale models, data were available for the external environmental climatological variables. To obtain this climatological base, we collaborated with the Spanish State Meteorological Agency (AEMET), who periodically provided us with all the data from the meteorological station closest to the scale models, i.e., the University City of Madrid (Spain).

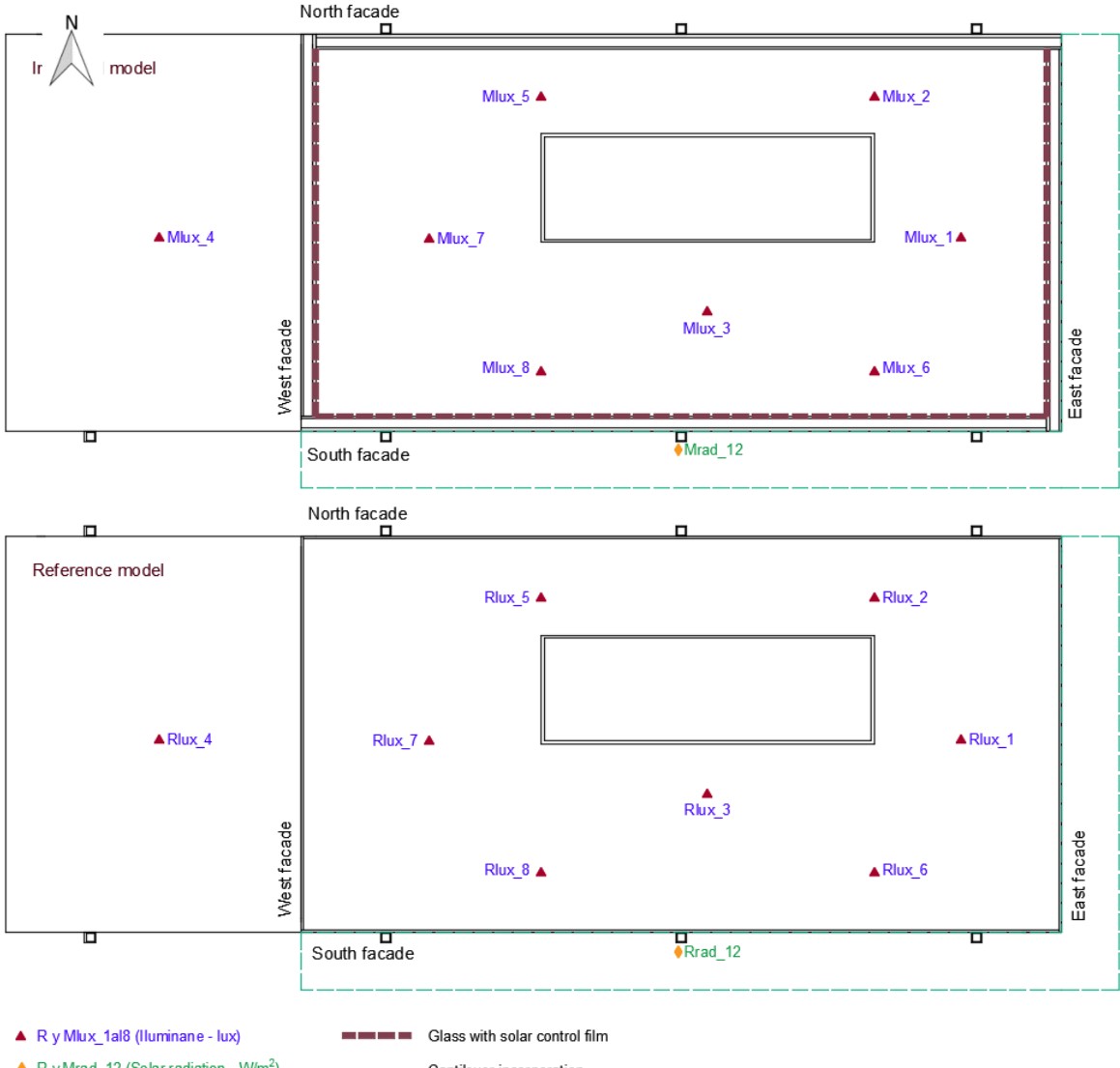

**Figure 3.** Position of the lux meters inside and the pyranometers outside the scale models.

### 2.3. Visual Comfort Light Parameters

In addition to the energy savings in artificial lighting demand brought about by daylighting, it is essential to take into account the well-being of occupants as a result of the right combination of quality and quantity of lighting. For this reason, the following parameters were also analyzed in this research:

- Uniformity and diversity.

In order to achieve a good lighting design, it is necessary that the rooms are illuminated as uniformly as possible. To calculate the uniformity (*U*), it is necessary to know the minimum illuminance of the room (*Emin*) and the average illuminance (*Em*) of the room under analysis, obtaining the result as a percentage.

$$U = Emin/Em \; (\%) \tag{1}$$

The uniformity values are included in the UNE EN-12464-1 (Spanish Association for Standardisation) for workplaces, but in residential use, no restriction is prescribed for the interior of dwellings. According to the bibliographies consulted, for residential use, the values recommended for the uniformity factor are between 40% and 70%, and this variable should increase with a higher percentage of the daylight factor; this will be analyzed later.

Another parameter to consider is the diversity factor (*D*), where the maximum illuminance of the room (*Emax*) is related to the minimum (*Emin*), obtaining the result as a percentage. The diversity should not exceed 5.

$$D = Emax/Emin \ (\%) \tag{2}$$

- Daylight factor

Illuminance and its distribution in the task area and in the surrounding area have a great impact on how a person perceives and performs a visual task quickly, safely, and comfortably. Therefore, this factor indicates the percentage of indoor illuminance in relation to the illuminance in an unobstructed horizontal outdoor plane under the same sky conditions. By considering cloudy sky conditions, direct radiation is excluded from the DF in the indoor and outdoor illuminance levels.

The design day for daylight factor calculations is based on the Commission Internationale de l'Eclairage or International Commission on Illumination (CIE) standard cloudy sky for 21 September at 12:00 h, where the terrestrial ambient light level is 11,921 lux.

The daylight factor at a point inside a room is the quotient between the illuminance at that point (*EI*) and the diffuse horizontal illuminance (*EH*), in percent.

$$FLD \ (\%) = EI/EH \times 100 \tag{3}$$

- Minimum and maximum illuminance.

Illuminance corresponds to the luminous flux received per unit area, expressed in $lm/m^2$. It is important to analyze the values that reach the minimum luminous flux on the work plane, as well as the maximum.

These values are legislated for workplaces according to the type and duration of the activity, as established by UNE EN 12464-1 and 2 (Spanish Association for Standardisation). This European standard specifies the lighting requirements for humans in indoor workplaces, in order to meet the comfort and visual performance needs of people with normal ophthalmological capacity. For residential use, there are no mandatory requirements in this respect; however, according to the publications consulted, the minimum illuminance recommended for indoor use in a home is above 200 lux, in order to be able to carry out a normal activity in the different rooms.

In addition, these maximum and minimum values indicate the number of hours of the day with solar radiation of a building which is both below and above the recommended illuminance, in order to be able to analyze the hours in which an additional supply of artificial lighting will be necessary, or the installation of some element of solar protection at specific times of the day. The more hours for which artificial lighting is needed, the greater the energy consumption in the building.

## 3. Results and Discussion of the Experimental Data

This section describes the results obtained from the lux meters and pyranometers installed on the two scale models, which collected data throughout the monitoring period spanning one full year. First, the data for the most representative period of each station were analyzed, followed by the illuminance by orientation, the visual comfort parameters, and finally the minimum illuminance in each station, which were studied seasonally to obtain an average value.

### 3.1. Experimental Data of Ambient Temperatures

Based on the seasonal statistical analysis throughout the monitoring year, the most stable period was chosen within each season. The summer parameters were recorded in two different years (2017 and 2018), and it was important that the external conditions during both selected periods were as similar as possible so that the samples were comparable. For this reason, the outdoor ambient temperature was compared during the summer season,

and we calculated the deviation in temperature between 2017 (where the scale models had no overhangs) and 2018 (where the scale models had overhangs). The temperature records were then averaged, and three consecutive days were found in which the average temperature in the period without a cantilever (2017) deviated by only 2% from that in the period with a cantilever (2018). In this way, it can be considered that the monitoring with and without a cantilever was carried out simultaneously. For the remaining stations, three consecutive days were selected where the average temperature difference for these three days was the lowest possible value from the average value calculated for each station [45]. The irradiance data were also taken into account in this analysis, verifying that, during the three selected days, the solar irradiance value experienced no alterations. Therefore, in the analysis of the behavior of the models, the three days chosen for each of the periods were named days 1, 2, and 3.

In the light analysis overthe summer season, it was observed in the scale model with cantilevers and with improved glass that the lighting level was reduced. In addition, it was observed that the overhangs produced a better light distribution, lowering the illuminance levels by around 12% compared to the models without overhangs. Dispersion graphs were also created, showing that, with low irradiance values, the models with overhangs experienced an increase in illuminance that did not exceed a value of 3000 lux. Moreover, in this case, there was less scatter in the indoor illuminance data. This shows that the incorporation of glass with a solar control sheet can reduce the levels of illuminance inside the scale models and that, in addition, the incorporation of cantilevers can produce a better distribution and control of light, attenuating the levels of luminous flux on the inside.

Figures 4 and 5 show the average of the indoor lux meters (green line for the reference model and red line for the modified model). The solar radiation (yellow line) shows the average of the two pyranometers outside the scale models.

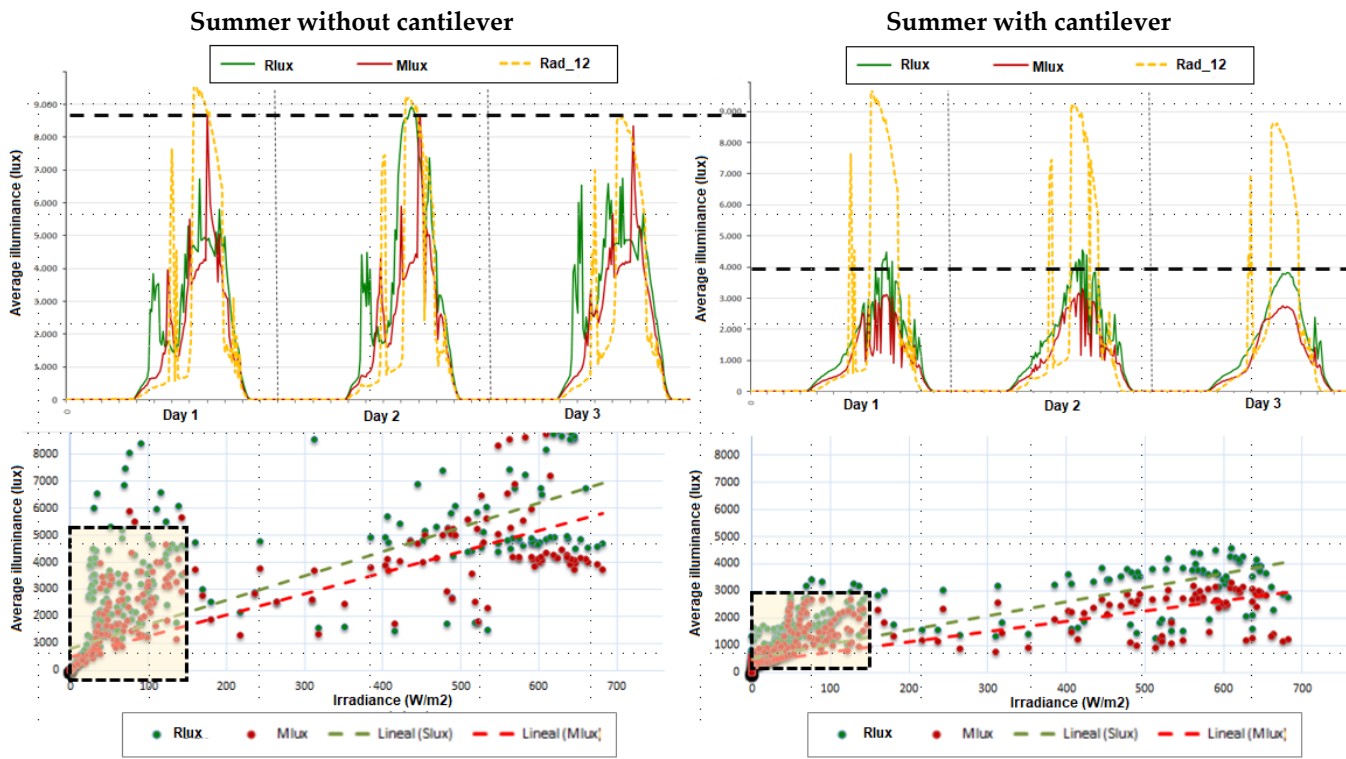

**Figure 4.** Average illuminance and irradiance in the summer season.

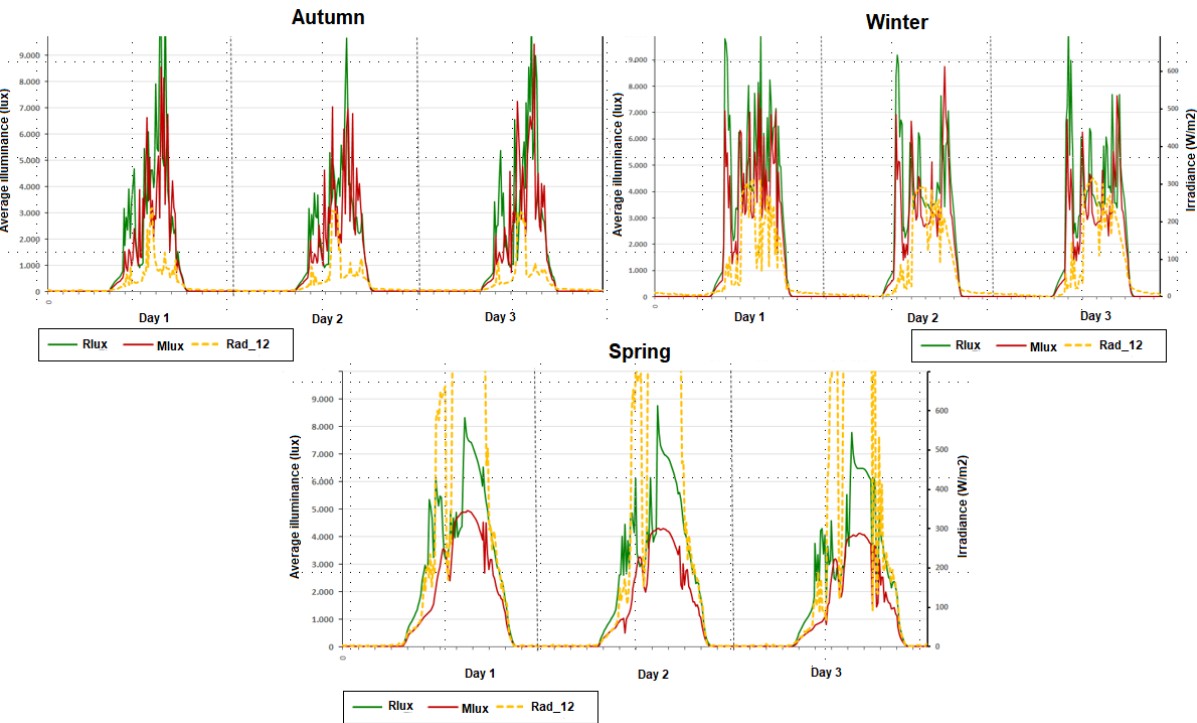

**Figure 5.** Average illuminance and irradiance in the autumn, winter, and spring seasons.

The incidence of radiation in the spring season with a lower solar altitude angle caused the illuminance inside to be distributed more evenly throughout the space. Additionally, the solar control sheet did not allow all of the incident solar radiation to enter compared to the remaining stations.

In autumn and winter (when there are fewer daylight hours per day), with less amplitude in the curvature of hours with solar radiation per day, the illuminance peaks were higher and more pronounced than those in spring (Figure 5).

### 3.2. Illuminance Analysis by Orientation

The following analysis shows the average illuminance values for each of the sensors, and it follows that, in summer, during the period in which passive protection was not available, the greatest difference between the improved model and the reference model was observed for the eastern orientation; however, with the installation of overhangs, significant differences—in addition to manifesting in this same orientation—were also reflected in the southern orientation. In the autumn season, the greatest difference was obtained for the lux meter located in the western orientation. Meanwhile, in winter and spring, the most significant differences were observed for the lux meter located in the eastern orientation (Figure 6).

In general, this shows that the incorporation of glass with a solar control sheet can reduce the levels of illuminance inside the scale models, with the eastern orientation being the one that obtains the best results.

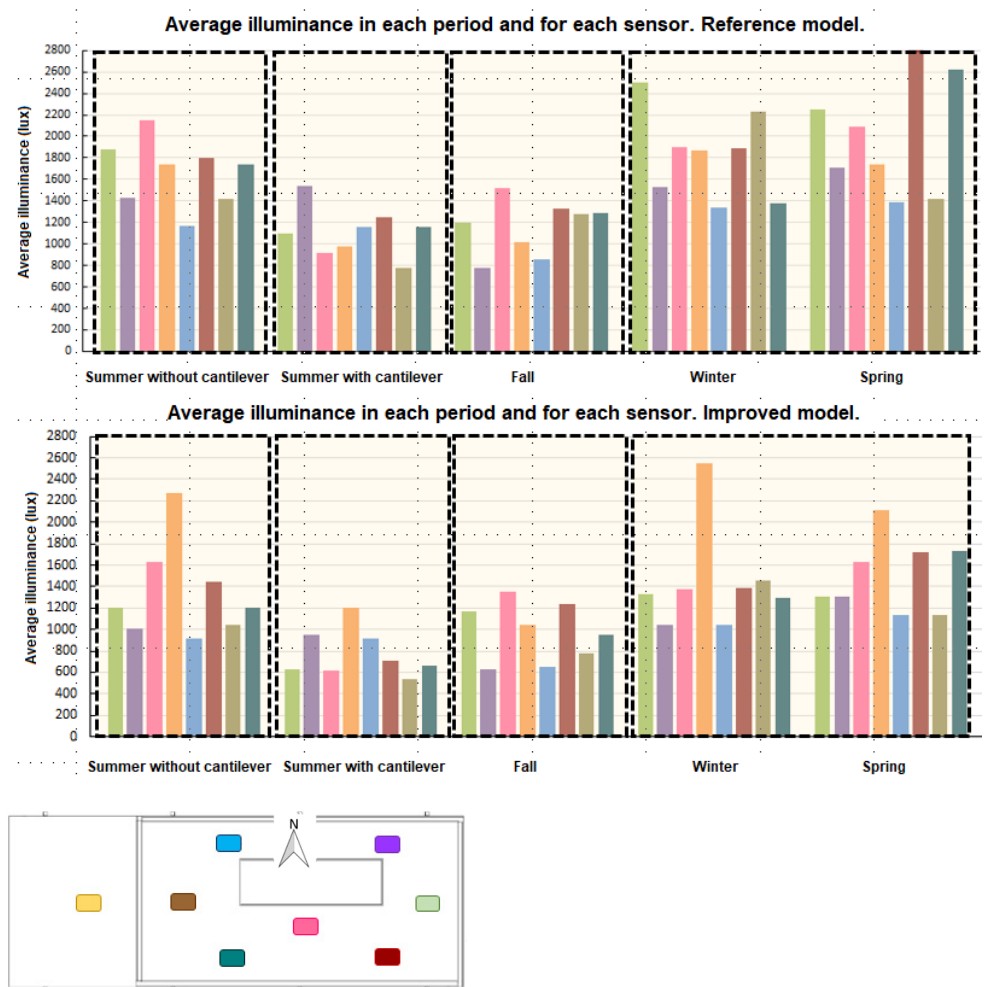

**Figure 6.** Average illuminance and irradiance in the autumn, winter, and spring seasons. Sensor positioning legend.

### 3.3. Visual Comfort Parameters

Regarding the visual comfort parameters, the uniformity was slightly higher in the simple glass model, with the exception of spring, where the value was observed to be in reverse. However, in all of the cases analyzed, the illuminance uniformity values were between 60% and 80%. On the contrary, the diversity should not exceed 5, with the highest value of all the data analyzed being 2.5.

Table 3 shows the results for each of the parameters analyzed for the reference model and the improved model. The percentage difference between the values of the two models was calculated, taking as a reference the value of the single glass model (reference).

The daylight factor for the improved glass model was 13.77%, while for the single glass model, it was 17.05%. The values were high, but since the uniformity values mentioned above were also high, visual comfort was compensated and suitable for indoor comfort levels.

For all of these reasons, the visual comfort parameters were considered to be acceptable in this typology of glazed dwellings for all of their façades and orientations.

**Table 3.** Uniformity and diversity of illuminance.

| | Mean Illuminance in the Horizontal Plane (lux) | Illuminance Uniformity (%) | Illuminance Diversity |
|---|---|---|---|
| | *Summer without Cantilever* | | |
| Reference | 1667.8 | 70.1 | 1.8 |
| Improvement | 1340.0 | 68.5 | 2.5 |
| Difference (%) | 19.65 | 2.28 | −38.89 |
| | *Summer with Cantilever* | | |
| Reference | 1107.2 | 70.0 | 2.0 |
| Improvement | 779.4 | 68.8 | 2.2 |
| Difference (%) | 29.61 | 1.71 | −10.00 |
| | *Autumn* | | |
| Reference | 1156.2 | 67.0 | 2.0 |
| Improvement | 976.4 | 64.4 | 2.2 |
| Difference (%) | 15.55 | 3.88 | −10.00 |
| | *Winter* | | |
| Reference | 1828.5 | 73.4 | 1.9 |
| Improvement | 1434.9 | 72.4 | 2.5 |
| Difference (%) | 21.53 | 1.36 | −31.58 |
| | *Spring* | | |
| Reference | 2017.8 | 68.7 | 2.1 |
| Improvement | 1510.6 | 75.0 | 1.9 |
| Difference (%) | 25.14 | −9.17 | 9.52 |

The percentage difference between the values of the two models is shown in blue. Uniformity is higher in the improved model (green) in the spring season. In the rest of the seasons it is lower (yellow).

### 3.4. Minimum Illuminance for Residential Use

It is noteworthy that there is no worldwide regulation that sets the minimum natural lighting parameters inside rooms, but according to the bibliographies consulted below, 200 lux is considered to result in insufficient vision to be able to carry out a task. Therefore, Figure 7 shows, in yellow, the hours of sunshine per day for each of the stations, while the green and red bars show the hours of natural light above 200 lux for each of the scale models.

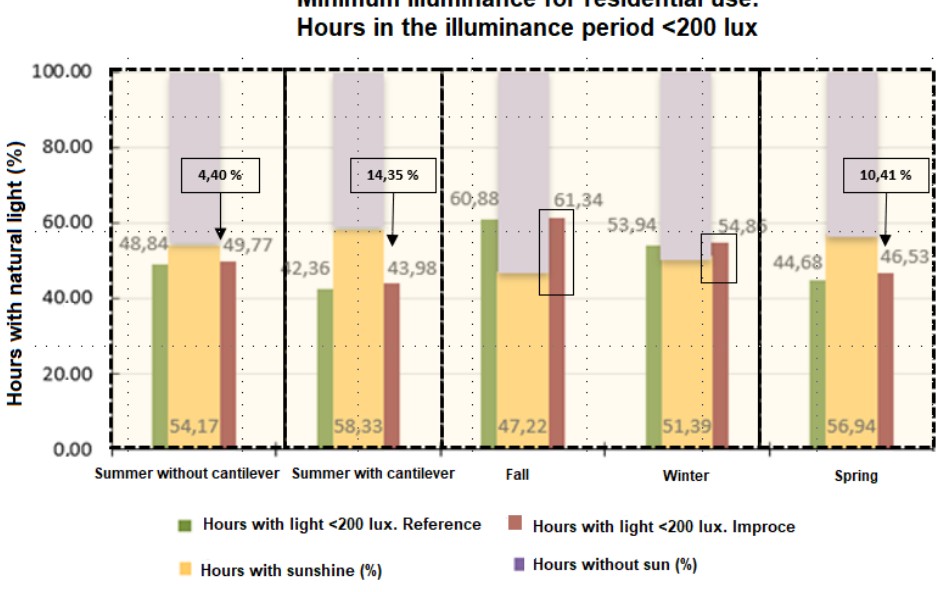

**Figure 7.** Hours of sunshine with illuminance <200 lux for each of the stations.

In the summer period without overhangs, both the improved and the reference models achieved very similar percentages, with the improved glass model with a solar control sheet being slightly higher and only 4.4% away from covering the total solar hours, which equates to 1 h per day.

In the summer period, with the installation of cantilevers facing south and east, it was observed that, in this case, the percentage of hours not covered in the improved glass model was 14.35%, approximately 3.5 h a day.

In the case of autumn and winter, the records collected for indoor illuminance showed that the percentage of hours above 200 lux was slightly higher than the hours of sunshine available during the day. In the spring season, 10.41% of the records were lower than 200 lux, which equates to 2.5 h a day.

From the above, it follows that only a minimal contribution of artificial lighting is required within the scale models.

## 4. Conclusions

The paper proposes a methodology which analyzes lighting parameters in dwellings with fully glazed façades. The analysis considers different scenarios, collecting data from two scale models with different passive strategies to be compared.

In particular, it was shown that the use of glass with a solar control sheet reduced the levels of illuminance inside the scale models and that, in addition, the incorporation of cantilevers produced a better distribution and control of light, attenuating the levels of luminous flux inside, with the eastern orientation being the one that obtained the best results for this type of glazed housing across all of its façades.

With respect to uniformity, it is concluded that, for all of the stations analyzed, the uniformity was slightly higher in the simple glass model, except for spring, where the value was observed to be in reverse.

The conclusions regarding the daylight factor indicated that the solar control foil on the glass had a positive influence. It was estimated that the daylight factor improved (decreased) by around 3.28% compared to the model without a solar control sheet.

It was found that, for most days of the year, for a climatic zone such as Madrid, an artificial lighting system is not necessary, thus minimizing energy consumption.

The energy improvement was located mainly in the field of the capture of solar energy, allowing its control and full use. The use of incident solar energy is essential to achieve maximum comfort in a building, with the minimum energy expenditure of non-renewable resources for artificial lighting. It should be noted that this study does not evaluate heating and cooling energy consumption, but rather how the use of passive strategies, such as solar control glazing and overhangs in dwellings with large glazed surfaces, affects natural lighting.

With the proposed approach, it is possible to transfer these criteria to the design of new buildings with large, glazed surfaces in their construction. The use of glass with a solar control sheet expands the freedom of architectural design, allowing the use of large, glazed surfaces, without compromising the energy efficiency of the building. This entails a new challenge when it comes to building houses with large, glazed surfaces, which, as was shown, are viable with respect to the lighting parameters, also allowing contact with the outside environment and permitting the entry of natural light, approaching regenerative construction. These are important and necessary characteristics in the context of the global pandemic caused by coronavirus disease 2019 (COVID-19), during which a large number of activities are being carried out inside the home.

**Author Contributions:** Conceptualization, P.A.-B. and S.V.-L.; methodology, P.A.-B.; software, C.P.-R.; validation, P.A.-B., S.V.-L. and C.P.-R.; formal analysis, P.A.-B.; investigation, P.A.-B.; resources, S.V.-L.; data curation, C.P.-R.; writing—original draft preparation, S.V.-L.; writing—review and editing, P.A.-B.; visualization, C.P.-R.; supervision, P.A.-B.; project administration, P.A.-B. All authors have read and agreed to the published version of the manuscript.

**Funding:** This research received no external funding.

**Institutional Review Board Statement:** Not applicable.

**Informed Consent Statement:** Not applicable.

**Data Availability Statement:** Not applicable.

**Conflicts of Interest:** The authors declare no conflict of interest.

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
