# Peer review of "Experimental Analysis of Passive Strategies in Houses with Glass Façades for the Use of Natural Light"

_sustainability, doi:10.3390/su13158652_

Round 1
Reviewer 1 Report
An overall convincing and interesting experiment. The methodology seems to be well set up and the testing of the full scale model is interesting. A key result seems to be about the benefits of solar control sheets.
It needs to be made clear as one of the limitations of this study that it does not evaluate energy efficiency of the Farnsworth House or glazed buildings generally in terms of total energy use during operation. The most significant factor in the energy use of a building like this would most likely be heating and cooling, not the energy used for artificial lighting. It needs to be made clear that this study does not evaluate those other aspects. It is focused on lighting and it therefore can't really claim that it is about overall energy efficiency.
Would it be possible to make it clearer in Figure 3 what exactly was changed in the improved model? Perhaps a call out with a note would help.
Style comments:
'Glass houses' seems to be a slightly problematic or misleading term. Glass houses is typically the term used for glass buildings in your garden to grow vegetables. Perhaps better: houses with glass facades. Or perhaps, it simply needs to be made clear in the title that the Farnsworth house is the specific case study used to assess the daylight performance of glass facades? Please check the English overall. There are quite a few sentences where the words are just not quite right, for example line 30: Currently, the construction sector generates a serious environmental impact ... Please change this to: ... significant environmental impactAuthor Response
Consulte el archivo adjunto. Gracias.

Reviewer 2 Report
The authors aim at studying the behavior of natural light in residential buildings characterized by glass facades. They carried out an experimental investigation of a scale model of Farnsworth House analyzed from August 2017 to July 2018. Illuminance and solar radiation were analyzed in the reference model and in the improved model, which was implemented with solar control sheets and by the addition of a cantilever. Orientation, size and location did not change.
Starting from the Literature review, which is not exhaustive, it is not clear where the novelty of this work is. Glass houses have been extensively investigated in the previous Literature and the effects of the passive strategies proposed by the authors are well known.
Finally, the authors state the importance of “regenerative sustainability” and “regenerative construction” (not clear) and the COVID-19 pandemic (also inserted in the keyword), which seems completely out of topic in a study focused on natural lighting in glass houses.
In conclusion, the paper as it is not suitable for publication.
Reviewer 3 Report
Dear Authors,
the problem of light in house is interesting research. Reducing impacts by maximizing the use of natural light is a good idea. However, the paper is a scientific article and to be published it is necessary to improve the quality.
Introduction: a good introduction was presented, but from my point of view, it is necessary to add additional studies or works of similar analysis. It is okay to indicate the problems of the glass industry, but what are the main causes of impact on the home?
For example, the following papers report a general assessment of domestic impacts:
Interactive energetic, environmental and economic analysis of renewable hybrid energy system, International Journal on Interactive Design and Manufacturing, 2019, 13(3), pp. 885–899
Ecodesign and Energy Labelling: The Role of Virtual Prototyping, Procedia CIRP, 2017, 61, pp. 87–92.
At the end of the introduction you could enter a description of the structure of the paper. (in the chapter 2 will be describe….)
Visual comfort parameters: I didn't understand how this parameter was calculated. More detail is necessary
Discussion: Before the conclusions I would have expected some discussions of the authors. What are the benefits? What are the limits? What are the difficulties? Is it an economically sustainable solution?
Reviewer 4 Report
The text shows an interesting approach on daylight performance in residential buildings. This is a field of interest, especially during the pandemic situation which has reduced the contact of people with daylight in their homes. However, I have some comments about this study which I have grouped in four points:
Thermal performance: the reduction of carbon dioxide emissions in buildings is linked, apart from artificial lighting energy consumption, to the energy demand for heating and cooling. Glazed buildings, especially in climates with a hot season, can be a problem in terms of overheating. More glass means more visual contact with the outside and more light, but also more heat loads due to the greenhouse effect. It should be noted.
Methodology: the methodology is partially explained. It lacks some details:
- the position of light meters and pyranometer (height with reference to the floor)
- the floor plan of the models with their orientation and surroundings (which can cause solar or light obstructions, as the pictures with vegetation show)
- identification of the measuring devices with the results (there is a confusion in the text and figures with Slux and Mlux)
- it is not clear if the values showed in figures 4 and 5 is the average between all sensors or the interior ones. If the exterior one is considered, it is misleading.
- The visual comfort parameters are not explained in the methodology. Which are the selected ones and how are they calculated. In the case of the daylight factor, has it been modelled or averaged with the sensors? Same consideration for <200lux hours.
Quality of figures:
- The scale of Figure 4 and 5 should be the same to compare them.
- Figures 4 and 5 seem incomplete and formally inconsistent between them (position of title, units, texts,..)
- Figure 6 should identify the colour of the bars with the number of sensors.
- Figure 7 is confusing. The percentage shown refers to the total hours with sunshine or the total of the day?
Significance of results: some conclusions are obvious because more glass implies higher illuminance values. However, there are some results that seem contradictory. For example, the difference between spring and autumn if the solar position is the same. Also, it is concluded that the glass with solar control offers a higher daylight factor than the glass without it. It would be advisable to explain why it happens.
Round 2
Reviewer 2 Report
The Authors have revised the paper with some additional information especially about the analyzed lighting parameters.
I appreciate the effort of the Authors, however I still have concerns about the novelty of the research topic which is focused on the study of well-known strategies (solar control sheets and a cantilever) and takes to mostly obvious results.
Reviewer 3 Report
Dear authors,
the paper has been improved in accordance with the suggestions of the reviewers. The state of the art is now complete and detailed.
The conclusions were extended and highlighted the main innovative aspects of the proposed method
Author Response
Por favor, vea el archivo adjunto.

Reviewer 4 Report
The context of the study is a broader work on energy efficiency of glass houses. It should be clear from the beginning of the text that this is a partial approach to energy consumption of such a typology. If it is not specified, it seems that the solution for zero energy buildings is to build glazed façades, when it is not probably so (considering the heating and cooling energy demand). Please, do not be so categorical with this assertion.
Other comments:
In Table 1, it is specified the shadow time for the reference and improved model, but the description of each type has not been made yet (it comes two paragraphs later).
With respect to the shadows on the models, it must be considered that the obstructions also apply to the sky vault. Daylight calculations consider the percentage of visible sky vault to determine the available daylight in a point. Therefore, sky obstructions count.
In line 160 specify if it is solar time or not.
When defining the diversity factor (D), please include the reference values. They appear in the results section.
In line 261, I think it is illuminance instead of luminance.
The experimental data of ambient temperatures is considered as a reference to choose the days of study. The values of solar irradiance would be more precise for this purpose, since they explain better the sky/solar situation in the period.
Figure 4 caption. It must be specified that the one on the left has no overhangs and the one on the right has them.
In the text, it is mentioned that autumn and winter have the same daylight hours per day, and it is not true. The solar position is different and also the duration of the day. Similar to this, I do not understand the different behaviour in autumn and spring in figure 5, considering that the solar condition should be comparable.
In line 325, it lacks one word defining the orientation.
Table 3 is not clear when defining the difference (%) between the reference model and the improved one. The percentage in blue corresponds to the increase-decrease of the mean illuminance value that appears in the table? Could you explain or check, please? (also correct the word AUTUM)
In the fourth paragraph of the conclusions, it is mentioned a positive influence and an improvement in the daylight factor. It means that it is reduced or increased?
Author Response
Por favor, vea el archivo adjunto.

Round 3
Reviewer 2 Report
Dear Authors,
thank you for the additional review and also for the extensive explanation of your work in your response.
Author Response
Thanks to you for your suggestions, which have improved the quality of the manuscript.